# Coupling Hydrogenation of Guaiacol with In Situ Hydrogen Production by Glycerol Aqueous Reforming over Ni/Al_2_O_3_ and Ni-X/Al_2_O_3_ (X = Cu, Mo, P) Catalysts

**DOI:** 10.3390/nano10071420

**Published:** 2020-07-21

**Authors:** Ziyin Chen, Roman G. Kukushkin, Petr M. Yeletsky, Andrey A. Saraev, Olga A. Bulavchenko, Marcos Millan

**Affiliations:** 1Department of Chemical Engineering, Imperial College London, London SW7 2AZ, UK; z.chen16@imperial.ac.uk; 2Federal Research Center, Boreskov Institute of Catalysis SB RAS, Lavrentieva Ave. 5, 630090 Novosibirsk, Russian; romang.kukushkin@yandex.ru (R.G.K.); yeletsky@catalysis.ru (P.M.Y.); asaraev@catalysis.ru (A.A.S.); isizy@catalysis.ru (O.A.B.)

**Keywords:** glycerol, guaiacol, Hydrogenation, In-situ generated hydrogen, Ni-based catalysts

## Abstract

Biomass-derived liquids, such as bio-oil obtained by fast pyrolysis, can be a valuable source of fuels and chemicals. However, these liquids have high oxygen and water content, needing further upgrading typically involving hydrotreating using H_2_ at high pressure and temperature. The harsh reaction conditions and use of expensive H_2_ have hindered the progress of this technology and led to the search for alternative processes. In this work, hydrogenation in aqueous phase is investigated using in-situ produced hydrogen from reforming of glycerol, a low-value by-product from biodiesel production, over Ni-based catalysts. Guaiacol was selected as a bio-oil model compound and high conversion (95%) to phenol and aromatic ring hydrogenation products was obtained over Ni/γ-Al_2_O_3_ at 250 °C and 2-h reaction time. Seventy percent selectivity to cyclohexanol and cyclohexanone was achieved at this condition. Hydrogenation capacity of P and Mo modified Ni/γ-Al_2_O_3_ was inhibited because more hydrogen undergoes methanation, while Cu showed a good performance in suppressing methane formation. These results demonstrate the feasibility of coupling aqueous phase reforming of glycerol with bio-oil hydrogenation, enabling the reaction to be carried out at lower temperatures and pressures and without the need for molecular H_2_.

## 1. Introduction

Biomass is considered a unique renewable carbon source for the production of fuels and chemicals. Bio-oil (or pyrolysis oil) can be directly produced from biomass via fast pyrolysis [1]. Nevertheless, low-quality bio-oil caused by high contents of oxygen and water makes it incompatible with petroleum-derived hydrocarbon fractions [2], while its low heating value, high viscosity, and corrosiveness limit its application as a source of renewable energy and chemicals [3]. Therefore, bio-oil requires upgrading in order to improve its fuel quality. Among the upgrading technologies, hydroprocessing, including hydrogenation and hydrodeoxygenation (HDO), has been proven effective to achieve a deoxygenated feedstock [4,5]. However, it involves reaction at 100 bar H_2_ pressure and temperatures of ~350 °C, which result in an expensive process for a relatively low-cost product such as fuel. A novel approach proposed to overcome these shortcomings is the use of in-situ generated hydrogen, which can be more reactive [6] and less costly compared to high-pressure molecular hydrogen.

Various alcohols and acids have been investigated as hydrogen precursors via aqueous phase reforming. An innovative depolymerization process of lignin was developed recently by the Weckhuysen group with a Pt/Al_2_O_3_ catalyst and using ethanol/water to produce in-situ hydrogen via aqueous phase reforming (APR) [7,8]. Total yield of monomeric phenolic compounds, including guaiacol and its derivatives, of 17% was obtained at 220 °C, and it was notable there was no char deposition observed after reaction. It was observed that the presence of ethanol has a positive effect on prevention of lignin repolymerization, while highly recalcitrant solids (char) are formed without ethanol. Similar approaches were followed in the study of hydrotreating of synthetic bio-oil with in situ-generated hydrogen over supported platinum catalysts [9], and the use of acids as hydrogen donors for upgrading bio-oil model compounds [10].

There are two prevalent types of C–O bonds in the phenolic compounds of bio-oils, hydroxyl (Csp_2_OH) and methoxy (Csp_2_OCH_3_). As guaiacol contains these two functional groups, and it is considered a representative bio-oil model compound [11]. Some studies in the literature have focused on guaiacol hydrotreating with in-situ generated hydrogen. Yu et al. [12] illustrated a new alternative for production of chemicals from the lignin depolymerization products via in-situ phenol or guaiacol hydrogenation at 220 °C over a Raney Ni catalyst using methanol as a hydrogen source. Nearly complete conversion of guaiacol was achieved after a 7 h reaction with over 90% selectivity towards cyclohexanol. Feng et al. [13] studied the in-situ hydrogenation process with the same conditions and catalyst. Apart from similar promising results obtained from hydrogenation of guaiacol, they found that the process could be utilized to improve efficiently the fuel quality of bio-oil and convert raw bio-oil into hydrocarbon transportation biofuel. A different hydrogen source (2-propanol) was tested in the hydrogenation/hydrogenolysis of guaiacol at 200 °C over a series of carbon-based catalysts [14].

Glycerol is an abundant by-product in biodiesel production while methanol and ethanol have better balance between supply and demand. To date, there are only few reports about utilization of glycerol as hydrogen source, which is abundant, renewable, and inexpensive [15]. Glycerol reforming in supercritical water was used as a hydrogen source for the in-situ reduction of graphene oxide [16]. Kim and co-workers [17] studied the application of glycerol as a hydrogen source through APR over Raney Ni to hydrogenate phenol, which was converted into cyclohexanone, cyclohexanol, benzene, and o-cresol.

Research studies carried out previously have shown that modification of Ni-based catalysts by Cu and/or Mo results in a significant increase in their activity in hydrogenolysis reaction via HDO of fast pyrolysis oil or its model oxygenates, including guaiacol [18,19,20] as well as others like esters of fatty acids [21,22,23]. Moreover, it was also found that the modification by Mo and/or P of NiCu HDO catalysts gives rise to enhancement of their corrosion resistance in the acidic medium of fast pyrolysis oil [24,25]. Thus, this study is focused on the hydrogenation/hydrodeoxygenation of guaiacol into cyclohexanol with in situ-generated hydrogen via glycerol aqueous phase reforming over a series of Ni-based alumina-supported catalysts promoted by Cu, Mo, and P.

## 2. Experimental

### 2.1. Reagents

Spherical γ-Al_2_O_3_ (Sasol, Germany) was used as catalyst support. Ni(NO_3_)_2_·6H_2_O, (NH_4_)_6_Mo_7_O_24_·4H_2_O, Cu(NO_3_)_2_·3H_2_O (analytical grade), H_3_PO_4_, and NH_4_OH were supplied by Reakhim Ltd. (Russia). Guaiacol (natural, ≥98%), and glycerol (BioXtra, ≥99%, GC) was obtained from Sigma Aldrich (Merck Life Science UK Limited, Gillingham, UK) in analytical purity.

### 2.2. Catalyst Preparation

All modified Ni-based catalysts were synthesized at the Boreskov Institute of Catalysis (SB, RAS). Nickel loading was kept constant at 20 wt.% for all catalysts in oxide form, and the loading of each metal promoter was maintained at 1 wt.% in oxide form (as CuO and MoO_3_).

The catalysts were synthesized in the following manner. The non-modified Ni/γ-Al_2_O_3_ sample was prepared via the incipient wetness impregnation of spherical γ-Al_2_O_3_ with nickel salt aqueous solution Ni(NO_3_)_2_·6H_2_O at 70 °C, followed by drying at 120 °C for 3 h and calcination at 550 °C for another 3 h in air. The copper-promoted Ni-Cu/γ-Al_2_O_3_ sample was prepared similarly, employing an impregnating aqueous solution of nickel and copper salts (Ni(NO_3_)_2_·6H_2_O and Cu(NO_3_)_2_·3H_2_O).

Ni-Mo/γ-Al_2_O_3_ and Ni-Cu-Mo/γ-Al_2_O_3_ were prepared via incipient wetness impregnation of Ni/γ-Al_2_O_3_ and Ni-Cu/γ-Al_2_O_3_ samples, respectively, by using an aqueous solution of (NH_4_)_6_Mo_7_O_24_·4H_2_O, followed by drying and calcination at the same conditions described above.

Ni-P/γ-Al_2_O_3_ was synthesized via incipient wetness impregnation of Ni/γ-Al_2_O_3_ with an aqueous solution of a mixture of ammonium phosphates obtained through neutralization of H_3_PO_4_ by NH_3_ water solution to pH 7 at room temperature with the appropriate phosphorus concentration to have a phosphorus loading of 1 wt.%. The same drying and calcination procedures were used.

### 2.3. Catalyst Characterization

X-ray diffraction (XRD) patterns of the catalysts were obtained using a Bruker D8 advance diffractometer (Bruker AXS LTD., Coventry, UK) with a θ–θ configuration using Cu Kα radiation at λ = 0.1542 nm. Scans were performed for 2θ values from 5° to 80°. Counts were accumulated every 0.05° at a step time of 120 s. The crystal size of a modified Ni catalyst was calculated using Scherrer’s equation.

Textural properties such as surface area and porosity of the Ni-based catalysts were analyzed through N2 adsorption in a Micromeritics Tristar 3000 analyzer (Micromeritics U.K. Ltd., Hertfordshire, UK). A sample of 100 mg of catalyst was degassed at 150 °C under N2 atmosphere in a Micrometrics Flow Prep 060 degasser. The degassed sample was then analyzed through a N_2_ adsorption standard test method at −196 °C. Results were obtained using the Brunauer–Emmett–Teller (BET) method to determine the surface area for adsorption incorporating multilayer coverage and the Barrett–Joyner–Halenda (BJH) method to calculate the pore size distribution in the catalyst from the experimental isotherms.

The elemental compositions of catalysts were analyzed by X-Ray Fluorescence spectroscopy (XRF) in an Epsilon 3XLE spectrometer (Malvern Panalytical Ltd., Malvern, UK), and the data were processed by using the Omnian software.

The reducibility of the catalysts was studied using temperature-programmed reduction (H_2_-TPR) analysis (Boreskov Institute of Catalysis, Russia). Two-hundred milligrams of sample was placed in a U-shaped quartz reactor and heated in a reducing atmosphere (20 mL min^−1^ of a mixture of 10% H_2_ and 90% Ar) at a constant heating rate of 6 °C min^−1^ to a temperature of 1000 °C. Changes in the hydrogen concentration at the reactor outlet were recorded using a thermal conductivity detector.

The morphology of fresh catalysts was studied by transmission electron microscopy (TEM) carried out on a JEM-2200FS (JEOL Ltd., Tokyo, Japan) microscope operating at an accelerating voltage of 200 kV. The lattice resolution of the devices is 1 Å on the grid. Instruments are equipped with energy-dispersive X-ray spectroscopy (EDX) prefixes for local elemental analysis. The JEM 2200FS has a scanning mode with a high-angle scattered electron detector (HAADF STEM method) and the ability to map the sample composition by element. Samples for electron microscopic studies were dispersed by ultrasound and applied by spraying ethyl alcohol on the substrate—mesh diameter of 3 mm, coated with a carbon film with a network of holes. To minimize the impact on the data of the EDX analysis, molybdenum grids and a beryllium holder were used.

X-ray photoelectron spectroscopy (XPS) was performed using an X-ray photoelectron spectrometer (SPECS Surface Nano Analysis GmbH, Germany) equipped with a hemispherical analyzer PHOIBOS-150, an X-ray monochromator FOCUS-500, and an X-ray source XR-50M (SPECS Surface Nano Analysis GmbH, Berlin, Germany) with a double Al/Ag anode. The XPS spectra were acquired using monochromatic Al Kα radiation (hν = 1486.74 eV) under ultra-high vacuum conditions. The binding energy (Eb) of the photoemission peaks were corrected to the Al 2p peak (Eb = 74.5 eV) of aluminium constituting the support γ-Al_2_O_3_. Relative concentrations of elements were determined from the integral intensities of the core-level spectra taking into account the photoionization of the corresponding terms. The spectra were analyzed using the CasaXPS software. A Shirley-type background was subtracted from each spectrum.

### 2.4. Catalytic Activity in the Hydrogenation of Guaiacol with In-Situ Glycerol Reforming as a Hydrogen Source

The layout of the stirred batch reactor employed in catalytic tests is shown in Figure 1. The reaction vessel had an inner diameter of 65 mm and a height of 55 mm, which gave a volume of 182 mL. The vessel was closed by a lid on which a K-type thermocouple (TC Direct 405-004), two needle valves (Swagelok 1/4 in), a pressure gauge (BOC 842346), and a magnetic drive (PARR A2140HC Series) were assembled. A further volume was related to the tubing between needle valves, the pressure gauge, and the lid, which gave a total volume of 190 mL. With the aim of ensuring safe operation at high pressure, the lid and walls of the vessel were 25 mm thick.

One-and-a-half grams of guaiacol and 75 mL 1 wt.% glycerol solution with 1 g Ni-X (X = Cu, Mo, P)/γ-Al_2_O_3_ were added to the reactor. Catalysts were reduced beforehand by exposing them to a hydrogen flow (30 mL min^−1^) at 650 °C for 2 h. The reactor was sealed and pressurized to 75 bar with N_2_ to check gas tightness using a leak detector. After the leakage test was completed, the reactor pressure was released and subsequently purged several times with helium and then pressurized again to 10 bar. It was then heated to reaction temperature (250 °C), and the magnetic stirrer was set to 400 rpm. As reaction time reached 2 h, the reactor was immersed rapidly into cold water in order to quench the reactions and left for 1 h to cool down the products to ambient temperature. Gas products were collected in a gas bag. The solids were separated from the mixture of water and liquid products by filtration using a Büchner funnel, flask and a silicon filter. Subsequently, the solids collected in a filter paper were then dried in a vacuum oven at 80 °C for 12 h and stored in a sample vial. The remaining liquid was extracted with chloroform three times in a separatory funnel at room temperature. The products soluble in chloroform were removed from the funnel and then the volume of the organic fraction was adjusted in a Buchi rotatory evaporator at 70 °C. Afterwards, the organic fraction was dried under N_2_ flow for 1 h to remove chloroform completely. Finally, the chloroform non-soluble products and water were stored in a sample bottle.

The calculation of guaiacol conversion as well as the yield of each product (cyclohexanol, cyclohexanone, phenol and 1-methyl-1,2-cyclohexanediol) were calculated by the following equations.
(1)$$\mathrm{Guaiacol}\;\mathrm{conversion}\;=\left(1-\frac{\mathrm{unreacted}\;\mathrm{guaiacol}\;}{\mathrm{input}\;\mathrm{guaiacol}\;}\right)\times 100$$
(2)$$\mathrm{Yield}\;\left(X\right)=\frac{\mathrm{Product}\;X\;\left(\mathrm{mol}\right)}{\mathrm{input}\;\mathrm{guaiacol}\;\left(\mathrm{mol}\right)}\times 100$$
(3)$$\mathrm{Selectivity}\;\left(X\right)=\frac{\mathrm{Product}\;X\;\left(\mathrm{mol}\right)}{\mathrm{Reacted}\;\mathrm{Guaiacol}\;\left(\mathrm{mol}\right)}\times 100$$
(4)$$H_{2}{\;\mathrm{selectivity}\;\mathrm{to}\;\mathrm{CH}}_{4}\;\left(\%\right)=\frac{H_{2}\;\mathrm{consumed}\;\mathrm{in}\;\mathrm{methanation}\;\left(\mathrm{mol}\right)}{{\mathrm{Total}\;H}_{2}\;\mathrm{produced}\;\left(\mathrm{mol}\right)}\times 100$$
(5)$$\mathrm{Coke}\;\mathrm{yield}\;\left(\%\right)=\frac{\mathrm{moles}\;\mathrm{of}\;C\;\mathrm{formed}}{\mathrm{moles}\;\mathrm{of}\;C\;\mathrm{in}\;\mathrm{feed}}\times 100$$

### 2.5. Gas Product Determination

The gas product was analyzed by gas chromatography (GC) in a PerkinElmer Clarus 500 (PerkinElmer Ltd., Llantrisant, UK) equipped with a thermal conductivity detector (TCD), and the gas product was separated in a capillary Carboxen 1010 Plot (30 m × 0.53 mm) column (Merck Life Science UK Limited, Gillingham, UK). A volume of 100 μL of gas sample was inserted into the injector kept at 100 °C. Helium was the carrier gas with a flow rate set at 4.0 mL min^−1^. The detector temperature was constant at 150 °C during the analysis. The GC temperature program was as follows; the oven temperature was kept constant at 30 °C for 5.5 min. Then, the temperature increased to 90 °C with a heating rate 20 °C min^−1^ and held at 90 °C for 2.5 min. Afterwards, the oven was heated up to 100 °C, held at that temperature for 3.5 min, and cooled down to room temperature when the analysis was finished.

### 2.6. Liquid Product Determination

Glycerol conversion was determined by high-performance liquid chromatography (HPLC) (PerkinElmer Ltd., Llantrisant, UK), with a BioRad HPX-87H column (Merck Life Science UK Limited, UK). The mobile phase was 0.005 M H_2_SO_4_ with a flowrate of 0.5 mL min^−1^ at 45 °C. A refractive index (RI) detector was employed. External calibration was conducted for quantification.

Guaiacol conversion and product distribution were determined by gas chromatography coupled with mass spectroscopy (GC–MS) (PerkinElmer Ltd., Llantrisant, UK) equipped with an Elite-5MS capillary column (L: 30 m, I.D.: 0.25 mm, film thickness: 0.25 mm) (Merck Life Science UK Limited, Gillingham, UK). The carrier gas used was helium. Prior to the analysis, a sample was prepared by dissolving 8.0 mg of chloroform-soluble products in 0.5 mL of chloroform. Then, a volume of 1.0 μL of the sample was injected. The temperature of injector was set at 280 °C and it operated in splitless mode. The oven temperature was kept constant at 50 °C for 5 min. Then, temperature was increased to 120 °C with a heating rate of 5 °C min^−1^ and held at this temperature for 5 min. After the analysis was completed, the oven was cooled down to 50 °C. Product identification was conducted by using the spectra library of National Institute of Standards and Technology (NIST) (Version 2.0 f, 2009) and a calibration curve based on guaiacol was made to quantify organic-soluble product.

### 2.7. Solid Product Determination

The amount of coke deposition was determined by using thermogravimetric analysis (TGA) in a PerkinElmer Pyris 1 analyzer (PerkinElmer Ltd., Llantrisant, UK). Ten milligrams of spent catalyst was used for the analysis. The TGA furnace temperature started at 30 °C with a holding time of 5 min and was then increased to 105 °C at 15 °C min^−1^. This temperature was held for 15 min with N_2_ as carrier gas to remove moisture completely in the catalyst. Subsequently, the carrier gas was switched to air and the furnace temperature increased to 800 °C at 10 °C min^−1^ and held for 10 min. Finally, the furnace was cooled down to ambient temperature.

## 3. Results and Discussion

### 3.1. Catalyst Characterization

Textural properties of fresh catalysts are summarized in Table 1. The BET surface areas of the fresh samples range between 111 and 162 m^2^ g^−1^. It was observed that the BET surface area of modified Ni/γ-Al_2_O_3_ catalysts was not affected by the additional Cu and/or Mo. However, significant changes in the surface area and pore volume of the P-modified catalyst were observed. Compared to Ni/γ-Al_2_O_3_, the BET surface area and pore volume of the Ni-P/γ-Al_2_O_3_ were lower while average pore diameter increased, which indicates the smaller alumina pores were partially blocked by phosphorus oxide species as well as Ni sites that could have redistributed during the modification.

XRD patterns of the fresh catalysts reduced at 700 °C are shown in Figure 2. The mean values of coherent scattering domain (CSD) and lattice parameters of Ni are summarized in Table 2. It is notable that the lattice parameter of metal of Cu-modified Ni catalysts is higher than that of the Ni/γ-Al_2_O_3_ sample, which indicates that a Ni-Cu alloy was formed. For the Ni-Cu/γ-Al_2_O_3_ sample, reflections of γ-Al_2_O_3_ (2θ = 32.3, 37.2, 39.7, 45.8, 66.9°) and metallic nickel (2θ = 44.7, 51.9, 76.3°) were observed, confirming that activation of the catalysts was effective. This is typical for NiCu catalysts [23,26]. Because of the mutual solubility of copper and nickel in the solid state, an isomorphous system can be formed regardless of composition [27]. Therefore, the absence of diffraction peaks correlated to metallic copper in the XRD pattern could be explained by the formation of this copper and nickel alloy.

Comparison between Ni/γ-Al_2_O_3_ and Ni-Mo/γ-Al_2_O_3_ catalysts can be performed based on data in Figure 2 and Table 2. The observed lattice parameter for Ni, a = 3.543 Å in Ni-Mo/γ-Al_2_O_3_, differed from the value for Ni/γ-Al_2_O_3_, where a = 3.523 Å [JCPD 04-0850], which may indicate Ni modification by Mo. Figure 3 shows TEM images of Ni/γ-Al_2_O_3_ and Ni-Mo/γ-Al_2_O_3_ samples reduced at 700 °C. It has been reported that the growth of metallic Ni particles is inhibited when Mo species are highly dispersed on γ-Al_2_O_3_ support [24] as a Ni–Mo solid solution or an intermetallic compound such as MoNi_4_ is formed and the growth of Ni particles is consequently suppressed. Although it was also observed [28] that the formation of merge phases could lead to the disappearance of diffraction peaks related to metallic Ni upon addition of Mo species, narrow diffraction peaks related to metallic Ni are still observed in the Mo-modified catalyst (Figure 2). Both these assumptions are in contradiction with our data probably because of different preparation way (Figure 2 and Figure 3). In our case, the particle size distribution for Ni and Ni–Mo samples are approximate, and the common NiMo phases were not detected, probably because of their X-ray amorphousness. As it could be seen from Figure 4 for Ni catalyst modified by Cu, the high-contrast regions on the HAADF-STEM images corresponds to the nickel and copper. The Ni–Cu catalysts, as it was mentioned above, were prepared through the co-impregnation way. Otherwise, in a case of the Mo- and P-modified catalysts, we could observe uniform distribution of Mo and P species over the catalyst surface.

The XRD pattern for Ni-Cu-Mo/γ-Al_2_O_3_ (Figure 2) showed wide peaks corresponding to the γ-Al_2_O_3_ support and the active component Ni. Ni lattice parameter (Table 2) for Mo-containing Ni catalysts was also higher than the Ni/γ-Al_2_O_3_ value.

From XRD data, we could not exclude the formation of NiO and NiAl_2_O_4_, as strong overlapping with the peaks of the support γ-Al_2_O_3_. In Ni-P/γ-Al_2_O_3_ the observed lattice parameter for Ni, a = 3.529 Å, is slightly higher than that for Ni in the Ni/γ-Al_2_O_3_ catalyst (Table 2). It is noteworthy that formation of AlPO_4_ can be produced by incorporation of P into the alumina support or the sequential impregnation of Ni and P precursors. However, it was not observed in the XRD pattern (Figure 2). Therefore, the absence of AlPO_4_ phase could be possibly explained by highly dispersed and amorphous AlPO_4_ species present in the catalysts [29].

The XRD patterns of used catalysts are presented in Figure 5. It is noticeable that a new crystalline phase, boehmite (AlOOH), was formed, showing main reflections 28.3°, 38.5°, 49.3°, 55.3°, 60.7°, 64.2° [JCPDS 21-1307] in all used catalysts except Ni-P/Al_2_O_3_, despite being reported [30,31] that the transformation of γ-Al_2_O_3_ into boehmite could be retarded in the presence of metal particles. It can be seen that the crystallinity of Ni/γ-Al_2_O_3_ and copper-modified Ni catalysts significantly increased, and the diffraction peaks related to metallic Ni were much sharper after the reaction, suggesting that the metal particles were potentially sintered. This is in line with evidence from Wen et al. [32] for spent Ni/Al_2_O_3_ catalysts after APR of glycerol at 230 °C and Freitas et al. [33] for Ni/Al_2_O_3_ catalyst after hydrogenolysis of glycerol with in-situ generated hydrogen. For Ni-Mo/Al_2_O_3_, Ni-Cu-Mo/Al_2_O_3_, and Ni-P/Al_2_O_3_ catalysts, diffraction peaks of γ-Al_2_O_3_ are observed.

Remarkably, it was found that there was no boehmite observed after reaction with Ni-P/γ-Al_2_O_3_ catalyst. In the case of Ni-P/γ-Al_2_O_3_, XRD pattern contains only diffraction lines of Ni and γ-Al_2_O_3_. As mentioned above, Ravenelle et al. reported that the transformation of γ-Al_2_O_3_ to boehmite could be affected through blocking of surface hydroxyl groups which served as initial hydration sites [30]. The availability of these surface hydroxyl groups was observed to decrease in a hydrotreating phosphorus-modified catalyst because of the formation of multiple bonding between the alumina support and phosphoric acid [34]. Therefore, boehmite formation was impeded with the addition of phosphorus due to reduction of amount of surface hydroxyl groups.

The Scherrer equation was used to calculate the average crystallite size and the peak used for calculation was that at 2θ = 44.5°, which accords with the (111) plane. The crystallite size of reduced catalysts was between 5.4 and 6.9 nm. Because of the additional Cu to the Ni catalysts, the crystallite size was slightly reduced from 5.7 to 5.4 nm for Ni/γ-Al_2_O_3_ and Ni-Cu/γ-Al_2_O_3_ samples, respectively. Cu addition resulting in crystallite size reduction was also observed in Ni-Cu catalysts prepared from hydrotalcite precursors tested in APR of glycerol [35] and Cu-Ni/γ-Al_2_O_3_ tested in hydrogenolysis of glycerol to propylene glycol [33].

It was found that crystallite size of metallic Ni increased from 5.7 to 6.9 nm with the addition of phosphorus to the catalyst. As a result of the phosphorus addition to Ni/γ-Al_2_O_3_, nickel species in nickel aluminate spinel structure were inclined to transform to nickel oxide phase. Compared to the nickel aluminate phase, the metal–support interaction of the nickel oxide phase is weaker [36]. Moreover, it is known that nickel oxide phase is more prone to sintering than the nickel aluminate phase at high temperature [37]. Therefore, it is believed that an increased ratio between nickel oxide and nickel aluminate phase, which may lead to sintering of nickel species, was responsible for an increment in metallic Ni crystallite size. Furthermore, it is noteworthy that crystallite size significantly increased after reaction, which suggests that sintering of these catalysts took place.

In order to study the fresh catalysts reducibility and the interaction between the metal species and γ-Al_2_O_3_, H_2_-TPR measurements were conducted and the obtained TPR profiles are presented in Figure 6. In the TPR profile of Ni/γ-Al_2_O_3_, a small shoulder at the lower temperature region of 250–500 °C and a wide peak with maximum at 691 °C in the high temperature region of 500–900 °C were observed. Well-dispersed NiO species, which demonstrate a weak interaction with alumina support, were reduced to Ni^0^ in the lower temperature region of 200–500 °C, while the maximum temperature of up to 900 °C is ascribed to the reduction of the non-stoichiometric nickel aluminate (NiO-Al_2_O_3_) and stoichiometric NiAl_2_O_4_ nickel aluminate. In this case, nickel particles having a stronger interaction with the support are reduced at a higher temperature [38].

The peak detected at ~750 °C in the TPR profile of Ni-P/Al_2_O_3_ is associated with reduction of nickel aluminate. It is notable that the profile of the reduction peak was more asymmetrical and the maximum of peak shifted to a higher temperature compared to the profile of Ni/Al_2_O_3_. It can be predicted that the asymmetrical change of the reduction peak was due to strongly enhanced interaction between nickel aluminate and support. According to previous studies [39,40], reduction of nickel aluminate with a high degree of nickel saturation is favorable rather than that with a low degree of nickel saturation, as less aluminium species surround nickel species. Therefore, it is believed that the increased reduction peak temperature in the TPR profile of Ni-P/γ-Al_2_O_3_ is caused by nickel aluminate spinel with lower degree of nickel saturation due to the addition of phosphorus to the support [36,41]. According to the literature, reduction of phosphorus species bound to oxygen takes place around 950–1000 °C [29,36]. The shoulder observed above 950 °C for Ni-P/γ-Al_2_O_3_ catalyst in Figure 6 is in good agreement with this.

The profile of Ni-Cu/γ-Al_2_O_3_ contains a distinct reduction peak at high temperatures (600–800 °C) with maxima at 690 °C, which relates to reduction of Ni aluminate species. Another small peak observed at 389 °C is ascribed to copper reduction from Cu^2+^ to Cu^0^ as the reduction temperature of pure CuO is in the region of 200–400 °C [42]. According to Carrero et al. [43], a synergistic interaction can take place when CuO is located close to other metal oxide phases that decreases the reduction temperature of the latter. Overall, the TPR profile of Ni-Cu/γ-Al_2_O_3_ is similar to that of Ni/γ-Al_2_O_3_, probably because of the small Cu content as shown in Table 3.

In the profiles of Mo-modified samples (Ni-Mo/γ-Al_2_O_3_ and Ni-Cu-Mo/γ-Al_2_O_3_), peaks observed around 660 °C (661 °C and 665 °C) can be attributed to the reduction of the nonstoichiometric nickel aluminate (NiO-Al_2_O_3_) [38]. The shoulder at 770 °C can be ascribed to the reduction of the octahedral Mo species which are agglomerated to the varying extent (from Mo^4+^ to Mo^0^), together with the partial reduction of tetrahedral Mo species interacting strongly with γ-Al_2_O_3_ [44,45].

For the copper-promoted sample Ni-Cu-Mo/γ-Al_2_O_3_, there is a notable small peak in the temperature range of 200 to 450 °C, attributed to the reduction of Cu^2+^ to Cu^0^ species.

Table 3 presents XPS data showing the relative atomic concentrations of elements in the catalysts as well as the binding energies of Al 2p, Ni 2p_3/2_, P 2p, Cu 2p_3/2_, and Mo 3d_5/2_. The Ni 2p_3/2_, Cu 2p_3/2_, and Mo 3d core-level spectra of the studied catalysts are presented in Figure 7; all spectra are normalized by the integral intensity of the corresponding Al 2p core-level spectra (not shown). It should be noted that Al 2p core-level spectra of the catalysts have a narrow symmetrical peak at 74.5 eV related to Al^3+^ in the Al_2_O_3_ support.

The Ni 2p_3/2_ core-level spectra have two peaks at 852.7 and 856.8–857.0 eV as well as corresponding satellite peaks. The peaks at 852.7 eV are attributed to metallic Ni. The detected Ni^0^ species may correspond to the reduction of NiO and Ni–O–Al which has a weak interaction with the support. Other peaks at 856.8–857.0 could be attributed to the Ni^2+^ species. These species were associated with low reducibility nonstoichiometric NiAl_x_O_y_ and stoichiometric NiAl_2_O_4_ formed due to strong interaction between Ni–O–Al and alumina [46]. It can be concluded that the surface Ni^2+^ species were not completely reduced to metallic Ni after the reduction treatment: approximately 10% of nickel species remained in Ni^2+^ state (in case of NiCuMo-catalyst the portion of non-reduced nickel is about 35%).

The P 2p core-level spectrum has one peak at 134.5 eV. According to the literature data, the peak from the range of 133.4-134.4 eV corresponds to phosphorus of (PO4)^3−^ phosphate groups [47,48].

The Cu 2p_3/2_ core-level spectra have one peak at 932.3 eV that can be attributed to Cu^1+^ and/or Cu^0^ states. According to the literature data, the Cu 2p_3/2_ binding energy for copper in the metallic state, and for Cu_2_O is in the range of 932.4 to 932.9 eV, while for CuO the Cu 2p_3/2_ binding energy is 933.6–934.6 eV [49,50,51,52]. Because the Cu 2p_3/2_ binding energies of Cu^1+^ and Cu^0^ states are similar, it hampers their identification. Nevertheless, the Auger-parameter α, which is equal to the sum of the Cu 2p_3/2_ peak binding energy and the Cu LMM peak kinetic energy, can be used for the identification of Cu^1+^ and Cu^0^ states. Due to low concentration of copper in these catalysts, determination of Auger parameter α characterizing an oxidation state of copper is impossible. However, previous studies of similar systems identified that copper was reduced to Cu^0^ state in catalysts after the hydrogen treatment at high temperatures and this could be the case.

The Mo 3d core-level spectra of studied catalysts have two Mo 3d_5/2_–Mo 3d_3/2_ doublets with Mo3d_5/2_ binding energies at 227.7 and 232.3–232.5 eV. The binding energy of Mo3d_3/2_ peaks of molybdenum in the state of Mo^0^, Mo^4+^, Mo^5+^, and Mo^6+^, according to published data is in the range of 227.6–228.0, 229.2–230.0, 230.8–231.6, and 232.3–233.0 eV, respectively [53,54]. According to literature data, the first asymmetric low-intensity doublet can be attributed to the Mo^0^ state or Mo^δ+^O_x_ species (0 < δ < 4) [55] and the second doublet is related to molybdenum in Mo^6+^ state.

Furthermore, it is notable that lower atomic ratios were measured by XPS than those calculated from XRF data, revealing the metal particles severely agglomerated during the reduction process while the non-metal particles (phosphorus) were not affected. This was confirmed by elemental mapping of the Ni-P/Al_2_O_3_ sample shown in Figure 4. The contrast areas on the HDAAF-STEM image showed particles with large Ni concentration while distribution of phosphorous seemed significantly more homogeneous.

### 3.2. In-Situ Hydrogenation of Guaiacol with the Ni-Based Alumina-Supported Catalysts

In order to investigate activity of catalysts in guaiacol hydrogenation in the presence of glycerol and water, all experiments were conducted at 250 °C using 1.5 g (12.1 mmol) of guaiacol in 75 mL of an aqueous solution of 1wt% glycerol (8 mmol) with 2 h reaction time. Ni/Al_2_O_3_, Ni-P/Al_2_O_3_, Ni-Cu/Al_2_O_3_, Ni-Mo/Al_2_O_3_, and Ni-Cu-Mo/Al_2_O_3_ were tested in the process. Glycerol conversion in all experiments was close to 100%. GC analysis identified H_2_, CO_2_, and CH_4_ as the main gas components with only traces of CO, as expected from conditions that favor the water gas shift reaction.

A high guaiacol conversion (95%) was achieved in the reaction catalyzed by Ni/Al_2_O_3_, as depicted in Figure 8. Its reaction products showed loss of the methoxy group and a high degree of hydrogenation, with cyclohexanol, cyclohexanone, and 1-methyl-1,2-cyclohexanediol accounting for 85% of the products (Table 4). No significant guaiacol reforming took place as can be seen from the high selectivities to organic-soluble products.

Based on the yields and product distributions obtained, the reaction pathway shown in Figure 9 is proposed. Guaiacol can undergo demethoxylation to form phenol followed by hydrogenation to produce cyclohexanone, which is subsequently reduced to cyclohexanol. An alternative but less favored pathway involves rearrangement to produce 1-methyl-1,2-cyclohexanediol without loss of oxygen [56]. Reports in the guaiacol HDO literature [57] postulate 1-methyl-1,2-cyclohexanediol as an intermediate in the reaction towards cyclohexanone and cyclohexanol, but results in Table 4 suggest that phenol is a more likely intermediate in this case as its selectivity is seen to decrease when catalytic activity is high. Full deoxygenation to cyclohexane was not observed.

Production of phenol, cyclohexanone, cyclohexanol, and 1-methyl-1,2 cyclohexanediol from guaiacol consumes 1 mol, 3 mol, 4 mol, and 3 mol H_2_ per mol of product, respectively. These stoichiometric relationships were used in conjunction with the product distributions to calculate the hydrogen produced in each experiment and consequently assess the performance of each catalyst. The theoretical maximum value of in-situ generated hydrogen is 87 mmol. This assumes complete reforming of glycerol according to C_3_H_8_O_3_ + 3 H_2_O → 3 CO_2_ + 7 H_2_, which would generate 57 mmol of H_2_, and the methoxy group in guaiacol according to -OCH_3_ + H_2_O → CO_2_ + 2.5 H_2_, which would contribute another 30 mmol.

The hydrogen balances are summarized in Table 5, which shows the total H_2_ produced and how it is determined as the sum of the net H_2_ production (hydrogen as H_2_ at the end of the reaction), H_2_ consumed in guaiacol hydrogenation to the various products observed, and H_2_ involved in CH_4_ formation. The main hydrogen-containing reaction products are H_2_, guaiacol hydrogenated products, and CH_4_. The latter is generated alongside H_2_O according to CO_2_ + 4 H_2_ → CH_4_ + 2 H_2_O, which allows H_2_ consumed in CH_4_ (and therefore total hydrogen production) to be calculated as shown in Table 5. It is worth noting that no mechanistic assumption is made regarding methane formation. The H_2_ used in producing CH_4_ may have been the result of CO or CO_2_ methanation reactions taking place but it is also possible that glycerol and/or the guaiacol methoxy group were transformed into CH_4_ by other routes. However, this does not affect the data presented in Table 5.

The amount of H_2_ produced in the run catalyzed by Ni/Al_2_O_3_ was higher than the one that would result from complete glycerol reforming. However, nearly 40% of H_2_ produced appeared as CH_4_, which competed with guaiacol hydrogenation.

All promoted catalysts showed less activity in comparison with Ni/Al_2_O_3_. Guaiacol conversion decreased from 95% to 50% (Figure 8) when phosphorus was used as a promoter and total yield of products (cyclohexanol, cyclohexanone, phenol, and 1-methyl-1,2-cyclohexanediol) was reduced from 85 mol% to 43 mol%. The selectivity towards cyclohexanol and cyclohexanone reached 45% with Ni-P/Al_2_O_3_ as opposed to 70% by using Ni/Al_2_O_3_. Guaiacol demethoxylation is the first step in the main reaction pathway and the significant drop in conversion observed is linked to this reaction being inhibited. The Lewis base properties of phosphorous [58,59] may have been affected this cracking reaction by decreasing the concentration of strong acid sites in γ-Al_2_O_3_ [34,58] even at the low concentrations employed [60]. Brønsted acid sites on γ-Al_2_O_3_, which play a role in promoting glycerol dehydration [61], may also have been affected by P addition as total hydrogen production dropped to 49.4 mmol, slightly below the theoretical maximum expected from glycerol reforming 57 mmol. On the other hand, the amount of CH_4_ formed increased despite the lower extent of demethoxylation, showing an enhanced pathway from glycerol to CH_4_. As highlighted in the previous section, the main advantage of the use of P was the preservation of the γ-Al_2_O_3_ structure following reaction.

Addition of copper as a promoter in Ni-Cu/Al_2_O_3_ led to a guaiacol conversion of 40% and total yield of products (cyclohexanol, cyclohexanone, phenol, and 1-methyl-1,2-cyclohexanediol) of 36 mol%. The selectivity towards aromatic ring hydrogenation was 51%, significantly lower than the 85% achieved with Ni/Al_2_O_3_, by contrast with the enhancement observed in HDO when Cu was added to Ni/δ-Al_2_O_3_ catalysts [62]. This drop in hydrogenation may be the result of the inhibition observed in total hydrogen production from glycerol reforming, which dropped to less than half that obtained with Ni/Al_2_O_3_. Adverse effect of Cu on glycerol reforming had been observed in Pt/γ-Al_2_O_3_ [63] and Ni/γ-Al_2_O_3_ [38] catalysts, but at much higher Cu:Ni ratio than used in this work. On the other hand, Cu had a positive effect by decreasing CH_4_ selectivity, albeit slightly, as formation of CH_4_ was less (2.8 mmol) with Ni-Cu/Al_2_O_3_ than Ni/Al_2_O_3_ (6.9 mmol).

In comparison with the catalytic performance of Ni/Al_2_O_3_, the addition of Mo had a negative effect on guaiacol conversion and total product yield, which declined sharply from 95% to 22% and 85 mol% to 16 mol%, respectively. The selectivity towards hydrogenation of the aromatic ring was 35%, below that obtained with Ni/Al_2_O_3_ and the P- and Cu-promoted catalysts. Similar to other promotors, Mo led to a poor activity in total hydrogen production from glycerol reforming, as shown in Table 5. This effect can be clearly seen in the comparison between. Both Ni-Cu/Al_2_O_3_ and Ni-Mo/Al_2_O_3_ showed similar total H_2_ production but while Cu tended to suppress methanation, Mo favored it. The use of Ni-Cu/Al_2_O_3_ therefore resulted in higher net H_2_ production and guaiacol hydrogenation but significantly lower selectivity to CH_4_ (36.5% vs. 68.9%) than Ni-Mo/Al_2_O_3_. It had been reported for a different type of catalyst (Pt supported on C [64]) that Mo decreased the activation energies for the cleavage of C–O bond in glycerol APR, favoring this pathway that leads to CH_4_ over dehydrogenation with H_2_ production. However, the reported increase in APR [64] rates achieved with Mo were not observed (Table 5).

Ni-Cu-Mo/Al_2_O_3_ was also tested with the same reaction conditions. Nevertheless, only 8.6% guaiacol conversion was achieved, and phenol was observed as a unique product possibly due to activity loss of catalysts as a result of the lowest accessibility of Ni active sites.

The coke yield in hydrogenation of guaiacol with in-situ generated hydrogen when using various modified Ni catalysts is presented in Table 6.

It is notable that the impact of different promoters on coke yield is small. Low coke yield was observed with all catalysts, and this is potentially caused by the low amount of carbon input and short reaction time.

## 4. Conclusions

A Ni/γ-Al_2_O_3_ and a series of promoted Ni-based catalysts (promoters: Cu, Mo, and P) prepared via the incipient wetness impregnation method were characterized with various techniques and applied to the hydrogenation of guaiacol with in-situ generated hydrogen from glycerol reforming. The feasibility of simultaneous H_2_ generation through glycerol aqueous phase reforming and hydrogenation of guaiacol was demonstrated with Ni/γ-Al_2_O_3_ leading to nearly total guaiacol conversion and high selectivity towards hydrogenation of the aromatic ring.

Overall, the type of promoters markedly affected the guaiacol conversion and the total yield of hydrogenated product. All three promoters have inhibitory effects on Ni catalytic activity even in the low concentrations used. The lower conversion and product yield obtained were mainly due to insufficient hydrogen, as the promoters decreased hydrogen production. However, methanation suppression by Cu is a beneficial effect, as it makes more H_2_ available for the hydrogenation of the aromatic ring, by contrast with Mo, which enhanced CH_4_ formation.

It was observed that changes in the catalyst structure took place after reaction, which could potentially lead to deactivation. Crystallite size increased significantly in all spent catalysts, which suggests sintering occurred, the reasons for which need to be further investigated. Boehmite formation was detected in the majority of spent catalyst supports, but was avoided by the addition of P in Ni-P/Al_2_O_3_, which makes it a promising catalyst in this sense.

Coupling aqueous phase glycerol reforming with in-situ hydrogen utilization avoids the need for molecular hydrogen to carry out hydrogenation reactions. Although at this stage this process needs further development to tackle some shortcomings of the catalysts, it can provide an attractive route for conversion of biomass-derived feedstocks such as lignin and pyrolysis oils.

## Figures and Tables

**Figure 1 nanomaterials-10-01420-f001:**
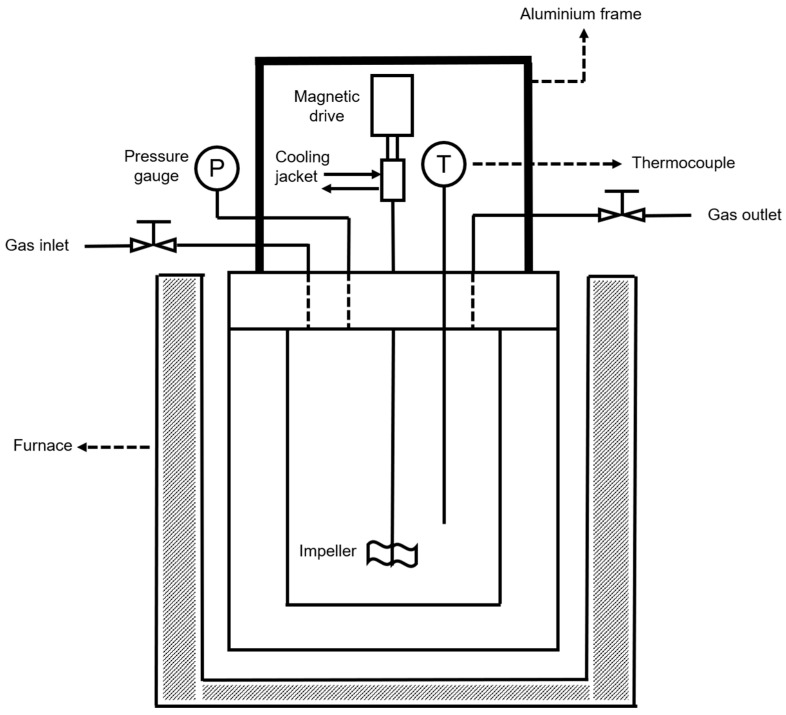
Layout of the stirred batch reactor.

**Figure 2 nanomaterials-10-01420-f002:**
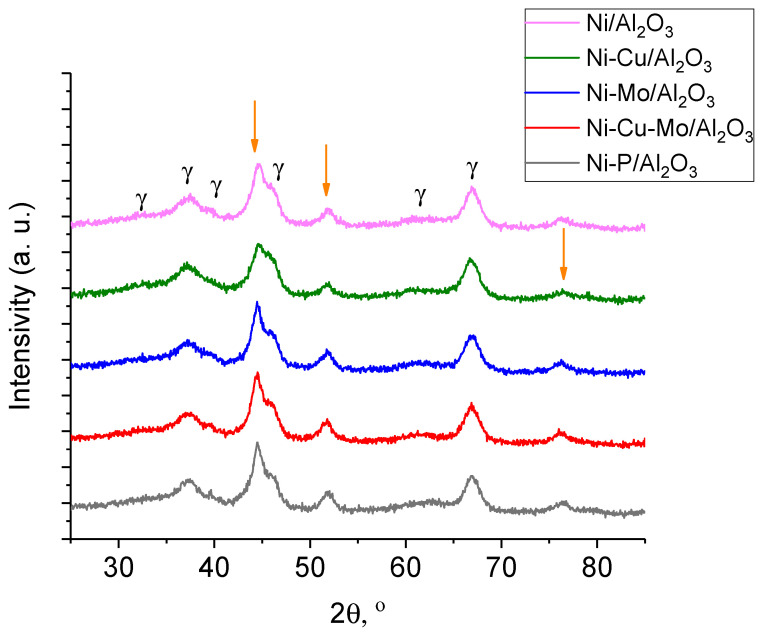
XRD patterns of fresh catalysts (reduced at 700 °C). Arrows indicate reflections of Ni, γ indicates γ-Al_2_O_3_.

**Figure 3 nanomaterials-10-01420-f003:**
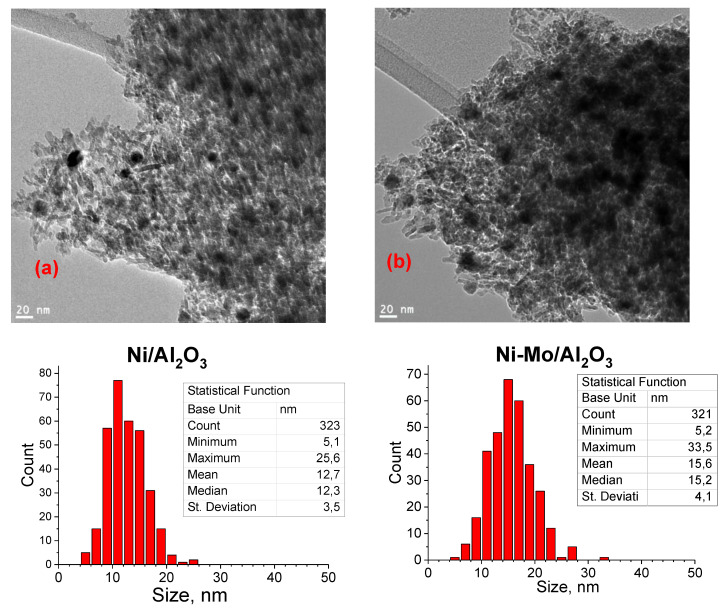
TEM images of (a) Ni/γ-Al_2_O_3_ and (b) Ni-Mo/γ-Al_2_O_3_ catalysts reduced at 700 °C and particles size distribution.

**Figure 4 nanomaterials-10-01420-f004:**
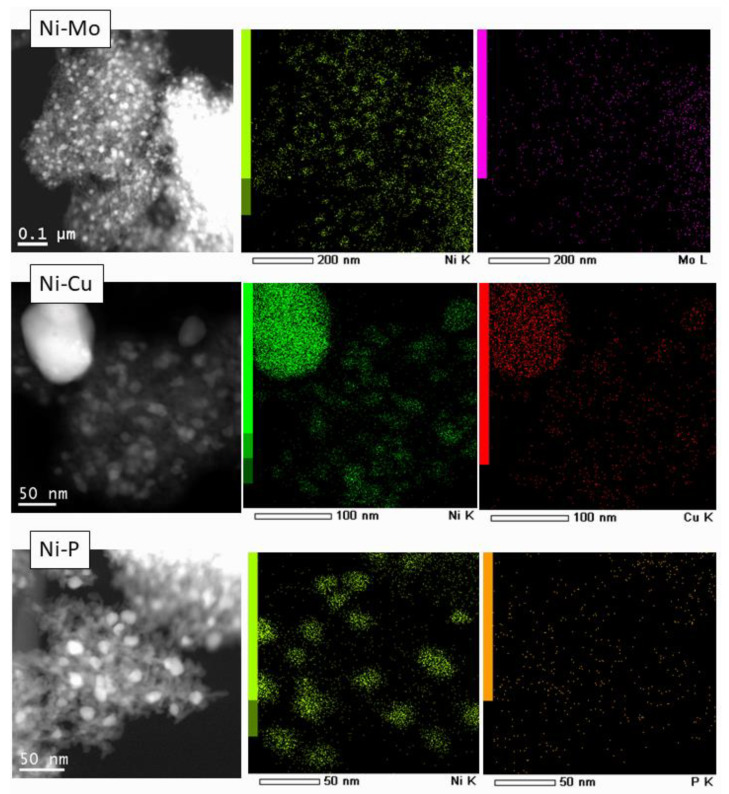
High-angle scattered electron detector (HAADF-STEM) images of Ni-P/γ-Al_2_O_3_, Ni-Mo/γ-Al_2_O_3_, Ni-Cu/γ-Al_2_O_3_, and elemental mapping (all samples reduced at 700 °C).

**Figure 5 nanomaterials-10-01420-f005:**
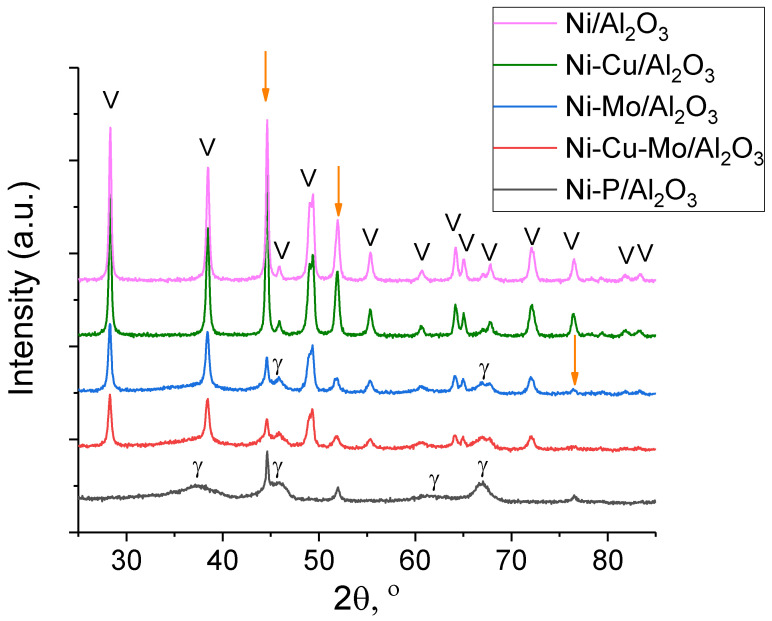
XRD patterns of used catalysts. Arrows indicate reflections of Ni, γ indicates γ-Al_2_O_3_, V indicates boehmite.

**Figure 6 nanomaterials-10-01420-f006:**
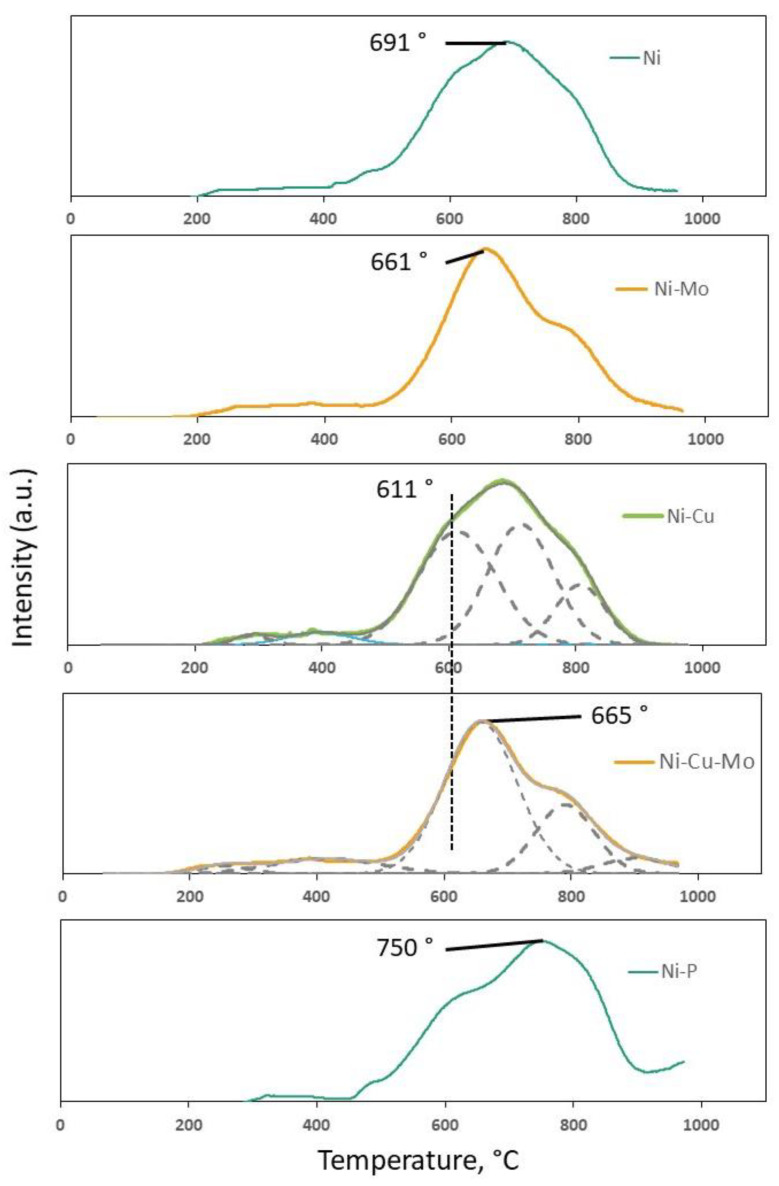
Temperature-programmed reduction (TPR) profiles of calcined Ni/γ-Al_2_O_3_, Ni-Cu/γ-Al_2_O_3_, Ni-Mo/γ-Al_2_O_3_, Ni-Cu-Mo/γ-Al_2_O_3_, and Ni-P/γ-Al_2_O_3_.

**Figure 7 nanomaterials-10-01420-f007:**
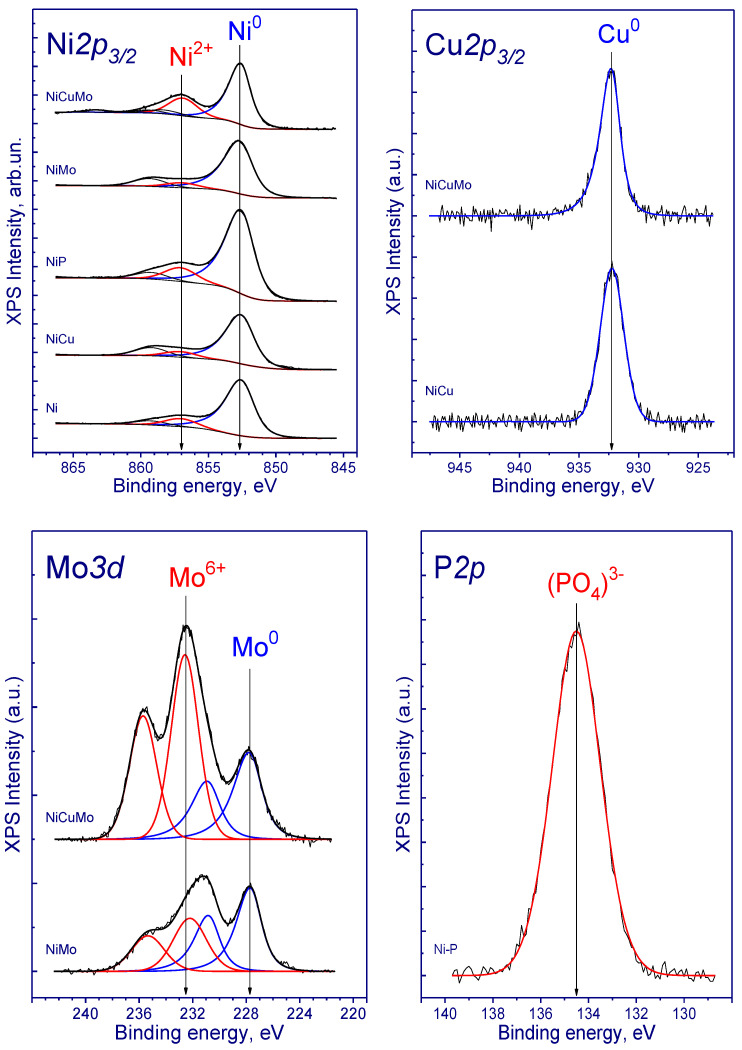
Ni 2p_3/2_, Cu 2p_3/2_, Mo 3d, and P 2p spectra of the studied catalysts (reduced at 700 °C).

**Figure 8 nanomaterials-10-01420-f008:**
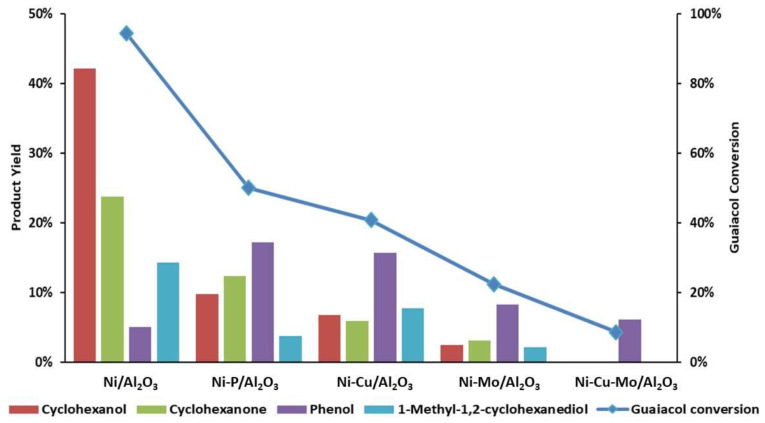
Hydrogenation of guaiacol with in-situ generated hydrogen at 250 °C by using 75 mL 1wt% glycerol under 10 bar initial pressure with 2 h reaction time.

**Figure 9 nanomaterials-10-01420-f009:**
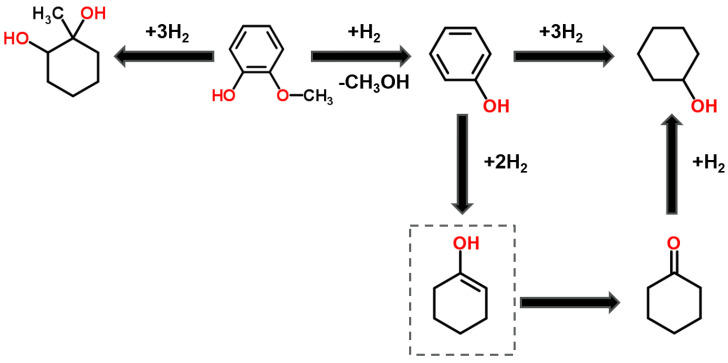
Proposed reaction pathway for the hydrogenation of guaiacol with in-situ generated hydrogen.

**Table 1 nanomaterials-10-01420-t001:** Textural properties (specific surface area, pore volume, and average pore diameter) of fresh catalysts.

Sample	S_BET_, m^2^/g	Pore Volume, cm^3^ g^−1^	Average Pore Diameter, nm
Ni/γ-Al_2_O_3_	150	0.36	9.7
Ni-Cu/γ-Al_2_O_3_	160	0.37	9.4
Ni-Mo/γ-Al_2_O_3_	162	0.38	9.3
Ni-Cu-Mo/γ-Al_2_O_3_	150	0.35	9.3
Ni-P/γ-Al_2_O_3_	111	0.28	10.2

**Table 2 nanomaterials-10-01420-t002:** Crystallite size of fresh and spent catalysts (coherent scattering domain), Å, Ni lattice parameter, Å for the model without NiO.

Sample	Crystallite Size of Ni for Fresh Catalysts, nm	Crystallite Size of Ni for Spent Catalysts, nm	Ni Lattice Parameter for Fresh Catalysts, Å
**Ni/γ-Al_2_O_3_**	5.7	40.6	3.523
**Ni-Cu/γ-Al_2_O_3_**	5.4	36.5	3.531
**Ni-Mo/γ-Al_2_O_3_**	6.8	20.9	3.543
**Ni-Cu-Mo/γ-Al_2_O_3_**	6.4	9.4	3.542
**Ni-P/γ-Al_2_O_3_**	6.9	35.9	3.529

**Table 3 nanomaterials-10-01420-t003:** Binding energies (eV) of Al 2p, Ni 2p_3/2_, Cu 2p_3/2_, Mo 3d_5/2_, and P 2p levels and surface atomic ratios.

Sample *	Al2p	Ni2p3/2		Cu2p3/2	Mo3d5/2		P2p	Ni/Al	Cu/Al	Mo/Al	P/Al
	Al_2_O_3_	Ni^0^	Ni^2+^		Mo^0^	Mo^n+^					
Ni/Al_2_O_3_	74.5	852.7	857.0	–	–	–	–	0.025 ^a^ 0.498 ^b^	–	–	–
Ni-P/Al_2_O_3_	74.5	852.7	857.0	–	–	–	134.5	0.038 ^a^ 0.574 ^b^	–	–	0.034 ^a^ 0.035 ^b^
Ni-Cu/Al_2_O_3_	74.5	852.7	857.0	932.3	–	–	–	0.023 ^a^ 0.474 ^b^	0.0056 ^a^ 0.0123 ^b^	–	–
Ni-Mo/ Al_2_O_3_	74.5	852.8	856.9	–	227.7	232.2	–	0.025 ^a^ 0.461 ^b^	–	0.0039 ^a^ 0.0343 ^b^	–
Ni-Cu-Mo/Al_2_O_3_	74.5	851.8	856.2	931.4	227.3	232.5	–	0.029 ^a^ 0.476 ^b^	0.0011 ^a^ 0.0166 ^b^	0.0066 ^a^ 0.0361 ^b^	–

* reduced at 700 °C. ^a, b^: atomic ratios calculated from XPS and X-ray fluorescence spectroscopy (XRF) data, respectively

**Table 4 nanomaterials-10-01420-t004:** Organic-soluble product selectivity of the experiments with modified Ni catalysts.

	Product Selectivity			
	Ni/Al_2_O_3_	Ni-P/Al_2_O_3_	Ni-Cu/Al_2_O_3_	Ni-Mo/Al_2_O_3_
Cyclohexanol	45%	20%	17%	11%
Cyclohexanone	25%	25%	15%	14%
Phenol	5%	34%	39%	37%
1-Methyl-1,2-cyclohexanediol	15%	8%	19%	10%

**Table 5 nanomaterials-10-01420-t005:** Hydrogen balance for hydrogenation of guaiacol with modified Ni catalysts and estimation of hydrogen production.

	Net H_2_ Production (mmol H_2_)	H_2_ consumed in Guaiacol Hydrogenation (mmol H_2_)	CH_4_ Production (mmol CH_4_)	H_2_ Consumed in CH_4_ Production (mmol H_2_)	Total H_2_ Production (mmol H_2_)	H_2_ Selectivity to CH_4_
Ni/Al_2_O_3_	7.5	34.6	6.9	27.6	69.7	39.6%
Ni-P/Al_2_O_3_	7.1	12.7	7.4	29.6	49.4	60.0%
Ni-Cu/Al_2_O_3_	9.1	10.0	2.8	11.2	30.3	36.5%
Ni-Mo/Al_2_O_3_	6.0	4.1	5.6	22.4	32.5	68.9%

**Table 6 nanomaterials-10-01420-t006:** Coke yields for hydrogenation of guaiacol with modified Ni catalysts.

	Ni/Al_2_O_3_	Ni-P/Al_2_O_3_	Ni-Cu/Al_2_O_3_	Ni-Mo/Al_2_O_3_	Ni-Cu-Mo/Al_2_O_3_
**Coke yield, wt.%**	7.7	6.8	7.6	7.1	7.0

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
