# Peer review of "Coupling Hydrogenation of Guaiacol with In Situ Hydrogen Production by Glycerol Aqueous Reforming over Ni/Al2O3 and Ni-X/Al2O3 (X = Cu, Mo, P) Catalysts"

_nanomaterials, 2020, doi:10.3390/nano10071420_

Round 1

Reviewer 1 Report

The present work of Millan et al. describes an elegant approach to hydrogenation of guaiacol using hydrogen obtained in situ by catalytic reforming of glycerol. In my opinion, such a method constitutes an original and economic approach to various derivatives of phenols and cyclohexanol. 

All the methods are clearly described and supported with transparent illustrations. Many catalysts modifications towards product selectivity are described which makes this work valuable from a synthetic point of view.

Taking all the notes into account, I hereby recommend this manuscript for publication in Nanomaterials in its current form. 

Author Response

Response to Reviewer 1 Comments

Point: The present work of Millan et al. describes an elegant approach to hydrogenation of guaiacol using hydrogen obtained in situ by catalytic reforming of glycerol. In my opinion, such a method constitutes an original and economic approach to various derivatives of phenols and cyclohexanol. 

All the methods are clearly described and supported with transparent illustrations. Many catalysts modifications towards product selectivity are described which makes this work valuable from a synthetic point of view.

Taking all the notes into account, I hereby recommend this manuscript for publication in Nanomaterials in its current form. 

Response: Thanks for the positive comments. We agree with this summary and the innovative process can provide an attractive route for conversion of pyrolysis oils.

Reviewer 2 Report

The present paper is very interesting and is worth to be published.

There are only few minor things to be corrected:

  1. In Fig 5, gamma is written two times: γ – γ-Al2O3.
  2. The figures shoudl be separated with a empty line to the text.
  3. At row 493 there is some yellow colour that has to be removed.

I kindly ask the authors to discuss the problem of hydrolization, taking into account that the catalytic support is alumina.

Author Response

Response to Reviewer 2 Comments

The present paper is very interesting and is worth to be published.

Point 1: In Fig 5, gamma is written two times: γ – γ-Al2O3.

Response 1: The first gamma is responded to the symbol γ which is displayed above the peaks in Figure 5 and the second γ is related to γ-Al2O3. Same as the caption in Figure 2.

Point 2: The figures should be separated with an empty line to the text.

Response 2: An empty line has been inserted between the figures and the text.

Point 3: At row 493 there is some yellow colour that has to be removed.

Response 3: The yellow colour has been removed.

Point 4: I kindly ask the authors to discuss the problem of hydrolyzation, taking into account that the catalytic support is alumina.

Response 4: Formation of boehmite was detected as shown in the Figure 5. In line 286, the reference 30 is cited. The transformation of γ-Al2O3 to boehmite is related to the surface hydroxyl groups, which act as hydration sites. Blocking surface hydroxyl groups is a potential way for stabilizing alumina support in hydrothermal reaction.

Reviewer 3 Report

This paper describes hydrogenation and hydrogenolysis of guaiacol to provide cyclohexanol,  cyclohexanone, and related compounds. Use of glycerol for hydrogen source is a characteristic aspect of this method over conventional methods, which can be interesting. This referee recommends publication of this work in Nanomaterials after suitable revisions.

  • Advantage of use of glycerol over methanol and ethanol should be discussed.
  • Which is preferable glycerol or ethanol/methanol from economical standpoint?
  • Formation of 1-methyl-1,2-cyclohexanediol is unusual. How was characterization conducted? What is the source of methyl group?

Author Response

Response to Reviewer 3 Comments

This paper describes hydrogenation and hydrogenolysis of guaiacol to provide cyclohexanol, cyclohexanone, and related compounds. Use of glycerol for hydrogen source is a characteristic aspect of this method over conventional methods, which can be interesting. This referee recommends publication of this work in Nanomaterials after suitable revisions.

Point 1: Advantage of use of glycerol over methanol and ethanol should be discussed.

Response 1: Firstly, the reports regarding to utilizing glycerol as hydrogen source are few which described in the introduction section (line 64-70). In addition, the price of glycerol is cheaper than methanol and ethanol.

Glycerol is abundant as by-product in biodiesel production while methanol and ethanol have better balance between supply and demand (inserted in line 64-65).

Point 2: Which is preferable glycerol or ethanol/methanol from economical standpoint?

Response 2: From economical standpoint, it is preferable to using glycerol as hydrogen source due to its lower price.

Point 3: Formation of 1-methyl-1,2-cyclohexanediol is unusual. How was characterization conducted? What is the source of methyl group?

Response 3: The organic-soluble products were characterized in a GC-MS and they were identified by using NIST library. According to the reference 56, 1-methyl-1,2-cyclohexanediol was formed in the HDO of guaiacol. The methyl group could derive from the methoxy group within guaiacol. The methyl group can be formed by rearrangement when guaiacol is converted to a substituted catechol structure.

Round 2

Reviewer 3 Report

3) Formation of 1-methyl-1,2-cyclohexanediol is unusual. How was characterization conducted? What is the source of methyl group?

Characterization of 1-methyl-1,2-cyclohexanediol is insufficient. Reference 56 tells “Due to the lack of reference reagents, the calibrations of trans and cis-1-methyl-1,2-cyclohexanediol were assumed to be the similar FID response as 2-methylcyclohexanol”. Authors need to verify the structure. This referee recommends publication of this work in Nanomaterials after suitable revision.
